# Mendelian randomization analysis to identify potential drug targets for osteoarthritis

**Chengyang Lu[1], Yanan Xu[2], Shuai Chen[1], Li Guo[1], Pengcui Li[1], Xiaochun Wei [1]\*, Xueqin Rong[3]\***

**1** Department of Orthopedics, Shanxi Key Laboratory of Bone and Soft Tissue Injury Repair, Second Hospital of Shanxi Medical University, Taiyuan, Shanxi, China, **2** Department of Laboratory, Second Hospital of Shanxi Medical University, Taiyuan, Shanxi, China, **3** Department of Pain Medicine Center, The Central Hospital of Sanya, Sanya City, Hainan Province, China

\* sdeygksys@163.com (XW); 13912048078@163.com (XR)

**Data Availability Statement:** The plasma pQTL data used for preliminary MR analysis in this study can be obtained from the deCODE Genetics website (https://www.decode.com/summarydata/). The

## Abstract

### Background

Osteoarthritis (OA) is a prevalent chronic joint disease for which there is a lack of effective treatments. In this study, we used Mendelian randomization analysis to identify circulating proteins that are causally associated with OA-related traits, providing important insights into potential drug targets for OA.

### Method

Causal associations between 1553 circulating proteins and five OA-related traits were assessed in large-scale two-sample MR analyses using Wald ratio or inverse variance weighting, and the results were corrected for Bonferroni. In addition, sensitivity analyses were performed to validate the reliability of the MR results, including reverse MR analysis and Steiger filtering to ensure the causal direction between circulating proteins and OA; Bayesian co-localization and phenotypic scanning were used to eliminate confounding effects and horizontal pleiotropy. External validation was performed to exclude incidental findings using novel plasma protein quantitative trait loci. Finally, the online analysis tool Enrichr was utilized to screen drugs and molecular docking was performed to predict binding modes and energies between proteins and drugs to identify the most stable and likely binding modes and drugs.

### Result

Four proteins were ultimately found to be reliably and causally associated with three OA-related features: DNAJB12 and USP8 were associated with knee OA, IL12B with spinal OA, and RGMB with thumb OA. The ORs for the above proteins were 1.51 (95% CI, 1.26–1.81), 1.72 (95% CI, 1.42–2.08), 0.87 (95% CI, 0.81–0.92), and 0.59 (95% CI, 0.47–0.75), respectively. Drug-predicting small molecules (doxazosin, XEN 103, and montelukast) that simultaneously target three proteins, DNAJB12, USP8, and IL12B, docked well.

plasma pQTL data used for external validation were obtained from Supplementary Table S1 in the original author's article (https://pmc.ncbi.nlm.nih.gov/articles/instance/7610464/bin/EMS118445-supplement-Supplementary_Tables.xlsx), after applying a selection process on Table S1. The GWAS data for osteoarthritis can be freely accessed through The Genetics of Osteoarthritis consortium (https://msk.hugeamp.org/downloads.html).

**Funding:** This manuscript was supported by funding from grants National Natural Science Foundation of China U23A6009, U21A20353, 82172503, Natural Science Foundation of Shanxi Province 20210302123285, Key R&D Program Projects of Shanxi Province 202202040201012, Hainan Provincial Medical and Health Research Program 21A200349. Dr. Xiaochun Wei, the principal investigator of the National Natural Science Foundation of China (grants U23A6009, U21A20353, and 82172503) and the Shanxi Key R&D Program (grant 202202040201012), provided valuable insights and revisions to the research content. the principal investigator of the Hainan Provincial Medical and Health Research Program (grant 21A200349), reviewed and edited the manuscript. Dr. Pengcui Li, the principal investigator of the Natural Science Foundation of Shanxi Province (grant 20210302123285), contributed to revising the framework and content of the manuscript.

**Competing interests:** The authors have declared that no competing interests exist.

## Conclusion

Based on our comprehensive analysis, we can draw the conclusion that there is a causal relationship between the genetic levels of *DNAJB12*, *USP8*, *IL12B*, and *RGMB* and the risk of respective OA.They may be potential options for OA screening and prevention in clinical practice. They can also serve as candidate molecules for future mechanism exploration and drug target selection.

## 1 Introduction

Osteoarthritis (OA) is the most common joint disease characterized by joint pain, swelling and dysfunction, which can lead to disability in severe cases [1]. The disease primarily affects the knee, followed by the lumbar spine, cervical spine, hands, ankles and hips [2]. The prevalence of OA is increasing every year as the population ages and the incidence of sports injuries increases, and the age of onset tends to be younger [3,4]. However, the pathogenesis of OA has not been fully elucidated and no drugs have been successfully developed to slow its progression [5,6]. Therefore, exploring the underlying mechanisms and drug targets of OA is of great value.

Proteins are arguably the ultimate players in all life processes in disease and health [7]. The human plasma proteome consists of proteins that are secreted or flow into the circulatory system, where they carry out their functions or mediate cross-tissue communication [8]. Dysregulation of the human plasma proteome is a common symptom of a wide range of diseases [9,10], as is the case with OA. Given the importance of circulating proteins, they are an attractive resource for finding drug targets for OA. At the same time, plasma samples are easier to collect and less invasive than other tissues, and data on genetic variation in plasma proteins are richer and more readily available, making them an important source for identifying molecular markers of disease in large cohorts.

Genome-wide association studies (GWAS) provide important molecular-level insights into the interplay of environmental and genetic factors in disease pathogenesis [11]. GWAS analysis of circulating protein levels identifies a large number of protein quantitative trait loci (pQTL) [12]. In addition, Mendelian randomisation (MR) analysis, which uses single nucleotide polymorphisms (SNPs) identified by global genome analysis as instrumental variables (IVs) to explore the causal relationship between exposure and outcome, is considered to be an effective method for assessing causality, where exposure refers to some environmental or biological factor of interest in the study [13,14]. MR analysis, which utilizes genotypes determined at the time of conception, effectively reduces the influence of confounding factors and thus clearly reveals causal effects [15]. Compared to traditional observational studies, MR analysis can provide more unbiased estimates and has been widely used to identify drug targets and repurpose existing drugs [16].

In our study, we used the MR framework to analyze the pQTL of circulating proteins in combination with SNPs for five OA-related traits (knee, hip, spine, finger, and thumb) to characterize the disease-causing effects of the proteins. This study is not only expected to deepen our understanding of the pathogenesis of OA, but also to provide potential pharmacological targets for subsequent therapeutic strategies. With this approach, we expect to provide more precise and effective scientific evidence for the prevention and treatment of OA.

## 2 Materials and methods

### 2.1 Study design

The use of SNPs as IVs in MR analyses to assess causal associations between exposures (e.g., circulating proteins) and outcomes (e.g., knee OA and hip OA) necessitates the fulfilment of three fundamental assumptions (Fig 1A). First, IVs must be demonstrably associated with the exposure; Second, IVs must not be associated with any confounding factors; Finally, IVs must influence outcomes exclusively through the exposure [17]. Based on these assumptions, the overall analytical framework is illustrated in the accompanying figure (Fig 1B). This causal analysis adheres to the Strengthening the Reporting of Observational Studies in Epidemiology (STROBE) guidelines, thereby enhancing the scientific rigor and standardization of the study (S1 Table) [18].

### 2.2 Source of data

Ferkingstad [19] et al.'s study measured the genotype data for 4,907 plasma protein quantitative trait loci (pQTL) in 35,559 Icelandic individuals, which currently represents a more comprehensive analysis of plasma proteins. In addition, plasma pQTL data obtained from a recent study by Zheng et al.[20] were used for external validation.

The summary statistics for OA were obtained from the largest GWAS meta-analysis published to date, which aggregated data from nine populations, involving 826,690 individuals (including 177,517 OA patients), and covering 11 OA-related traits [21]. In this study, we selected five specific OA sites for analysis: knee OA, hip OA, spine OA, finger OA, and thumb OA (Table 1).

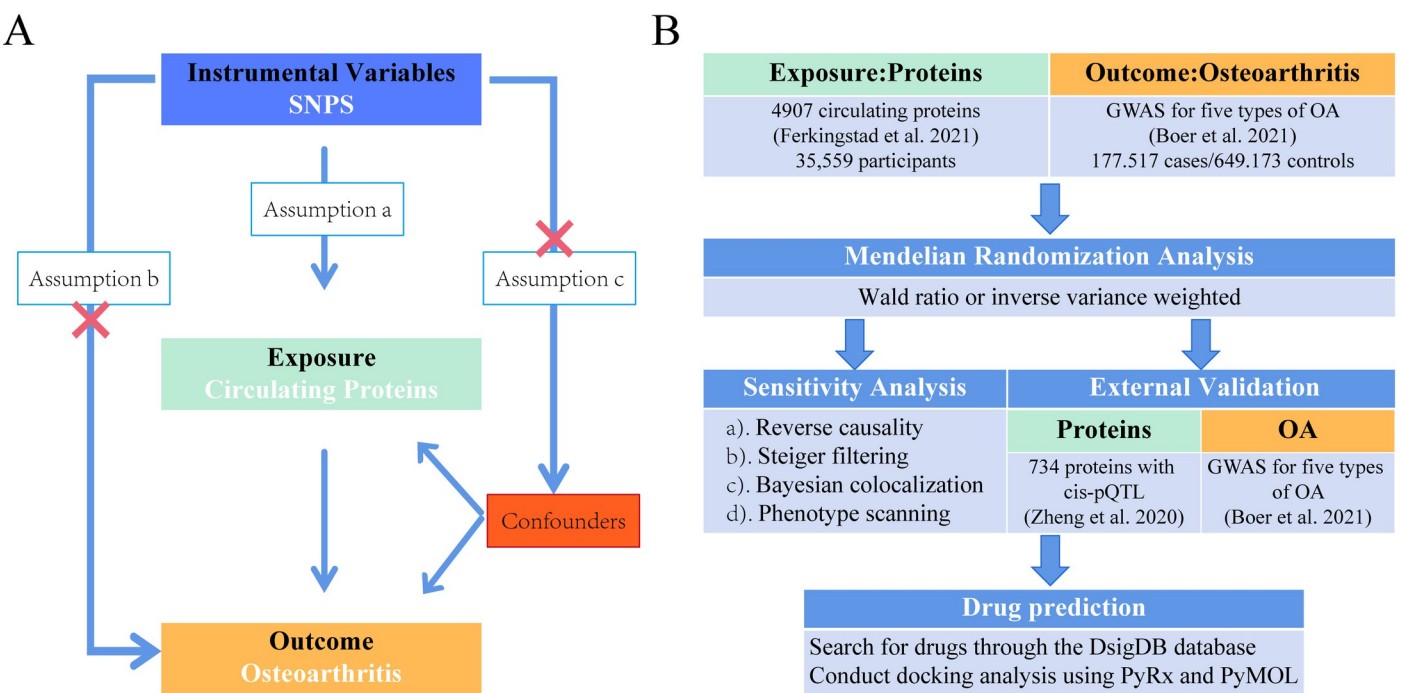

**Fig 1. Schematic representation of the study design.** (A) The figure depicts the three key assumptions in the MR analysis. (B) Overall analysis process.

**Table 1. Detailed information of the genome-wide association study of OA.**

| Phenotype | Data sourse | No. of cases | No. of controls | Sample size | Population |
|---|---|---|---|---|---|
| Knee OA | 9 cohorts of European and 1 cohorts of East Asian | 62,497 | 333,557 | 396,054 | European (99%) |
| Hip OA | 9 cohorts of European | 36,445 | 316,943 | 353,388 | European |
| Spine OA | 7 cohorts of European and 2 cohorts of East Asian | 28,372 | 305,578 | 333,950 | European (98%) |
| Finger OA | 6 cohorts of European | 10,804 | 282,881 | 255,814 | European |
| Thumb OA | 6 cohorts of European | 10,536 | 236,919 | 247,455 | European |

## 2.3 IVs selection

Firstly, this study selected cis-pQTLs related to circulating proteins (defined as SNPs located within a 500 kb window on either side of the leading pQTL for a specific protein, which are more likely to directly influence the corresponding protein's expression and thereby reduce the risk of pleiotropy and confounding effects compared to trans-pQTLs) [22], with a threshold of $P < 5 \times 10^{-8}$ to satisfy the correlation assumption [23]. Secondly, to address the issue of linkage disequilibrium, criteria of $r^2 < 0.001$ and a window size of 10,000 kb were employed to ensure the independence of SNPs [24]. Subsequently, palindromic sequences were removed using the harmonise_data() function from the TwoSampleMR package(https://github.com/MRCIEU/TwoSampleMR), thereby ensuring the reliability and accuracy of the genetic variants [25].

## 2.4 Mendelian randomization analysis

Plasma pQTL and five specific outcomes were also analysed by MR in this study using the 'TwoSampleMR' toolkit. When only one pQTL is available for a specific protein, the Wald ratio method is employed to estimate the causal effect [26,27]. The Wald ratio is appropriate in this case because it directly divides the SNP-outcome association by the SNP-exposure association, providing a straightforward and reliable estimate when a single instrumental variable is used. However, if two or more instrumental variables are available, the inverse variance-weighted Mendelian randomization (IVW) method is applied [28]. The IVW method combines estimates from multiple SNPs using a meta-analysis approach, assuming all SNPs are valid instruments. This increases statistical power and improves precision by pooling the information from multiple independent variants [28]. The results were measured by an increase in standard deviation (SD) of plasma protein levels, and were expressed as odds ratios (OR) along with a 95% confidence interval(CI). In the preliminary analysis, Bonferroni correction was used for multiple testing adjustment, and a threshold pvalue (P < 0.05/number of proteins) was applied for result selection and further analysis.

## 2.5 Reverse causality test

Based on the previously mentioned significant SNP criteria, we conducted a repeat MR analysis by swapping the exposure and outcome variables from the preliminary analysis. For this analysis, we employed the IVW, MR-Egger [29], weighted median method [30], simple mode, and weighted mode [31] to compute the results. In cases where these methods yielded inconsistent results, we used the IVW method as the reference standard. The IVW method was chosen because it accounts for heterogeneity in variance-specific causal estimates and combines ratio estimates through meta-analysis [28]. It ensures the reliability of the results under the assumption that all selected instrumental variables are valid, especially in the absence of horizontal pleiotropy [32]. If the pvalue is greater than 0.05, it indicates no causal association between the exposure factor and the outcome. Furthermore, we performed Steiger filtering as a dual test to

ensure the causal direction between circulating proteins and OA [33], and to prevent issues where significant SNPs for OA could not be matched in the corresponding protein data during reverse MR analysis. Results with a pvalue < 0.05 were considered statistically significant.

## 2.6 Bayesian colocalization analysis

We used the "coloc" package (https://github.com/chr1swallace/coloc) with default parameters for conducting Bayesian colocalization analysis to test whether two phenotypes share the same causal variant within a specific region. This analysis provides posterior probabilities for five hypotheses concerning whether the two traits share a single causal variant[34]. Among the five hypotheses, we focused on hypothesis 4 (PPH 4), which posits that "phenotype 1 and phenotype 2 are significantly associated with an SNP locus in a specific genomic region and are driven by the same causal variant." Specifically, we aimed to determine whether the association between the protein and OA is driven by the same causal variant. In this analysis, we defined the variant region as ±500 kb around the significant plasma pQTL locus and used the coloc.abf algorithm for colocalization analysis. We set the posterior probability threshold to be greater than 0.8 as the criterion for successful colocalization [20,35].

## 2.7 Phenotype scanning

We also used the tool "phenoscanner" [36] to perform phenotype scanning by searching previous GWAS, revealing the associations between identified plasma pQTLs and other traits. For SNPs considered to be pleiotropic, we followed the following criteria: (i) significant association across the entire genome ($P < 5 \times 10^{-8}$); (ii) GWAS conducted in populations of European ancestry; (iii) SNP associated with any known risk factors for OA, including metabolic characteristics, protein, or clinical features.

## 2.8 External validation

We utilized the plasma pQTL data from Zheng et al. [20], encompassing 738 cis-acting SNPs for 734 proteins, as new endogenous exposures. We conducted MR analysis on aggregated data for the same five OA phenotypes. Similarly, we applied Bonferroni correction to adjust for multiple testing, setting the significance threshold at 0.05/734 ($P < 6.81 \times 10^{-5}$).

## 2.9 Assessment of sample overlap using LD score regression

To assess the genetic correlation between proteins and OA, we employed the cross-trait LDSC method. This method, proposed by Bulik-Sullivan et al. [37], allows for the evaluation of the genetic correlation between two phenotypes and the detection of sample overlap. In cross-trait LDSC regression, the intercept is a key metric used to assess the extent of sample overlap. If there is no sample overlap between the protein and OA phenotypes, the estimated intercept value should be close to zero. Conversely, if the intercept significantly deviates from zero, it indicates that sample overlap may exist and could affect the genetic correlation estimates. The genetic correlation (rg) is calculated through the regression slope and represents the degree of shared genetic variation between the two phenotypes. Since the genetic correlation estimate is not influenced by sample overlap, it provides an accurate reflection of the genetic relationship between proteins and OA. Additionally, cross-trait LDSC analysis can help identify potential biases arising from sample overlap or LD reference mismatch.

In this study, we used the R package ldscr to perform cross-trait LDSC analysis, calculating the genetic correlation and intercept values between the major proteins and OA. This method

enables us to effectively assess the genetic relationship between proteins and OA, while also exploring the potential impact of sample overlap on the results.

## 2.10 Drug screening and molecular docking

After a series of validations, we obtained the 3D structures of the target proteins from the RCSB Protein Data Bank database [38], and the 3D structures were selected based on the principle that the structures should be as complete as possible with low conformational resolution. We then used the DsigDB database to find the small molecule drugs that act on these target proteins, and downloaded the corresponding 3D structures of these drugs via PubChem. We then used PyRx to dock the 3D structures of these proteins and small molecules and calculate their binding energies. In general, the lower the conformational stability energy of a ligand and receptor, the more likely they are to interact. Since there is no common standard for drug target screening, we chose drugs with binding energies below -6.9 kJ/mol as the basis for screening and used PyMOL to visualise the ligand-receptor with the best binding.

# 3 Result

## 3.1 Screening for significant IVs

After selecting for genome-wide independence ($r^2 < 0.001$, window size = 10000 kb) and $P < 5 \times 10^{-8}$, 1858 significant SNPs for 1553 proteins were screened; in the reverse MR analyses, the significant SNPs for the five OA's ranged from 1–38 (S2 Table).

## 3.2 Causal proteins in five types of OA

According to the Bonferroni-corrected threshold ($P < 0.05/1553 = 3.22 \times 10^{-5}$), the MR analysis revealed a total of 10 proteins (Tables 2 and S3). Within Knee OA, three proteins were identified: *DnaJ homolog subfamily B member 12* (*DNAJB12*), *Ubiquitin carboxyl-terminal hydrolase 8* (*USP8*), and *Collagen alpha-1(II) chain* (*COL2A1*). In Hip OA, three proteins were found: *Integrin alpha-2* (*ITGA2*), *Hedgehog-interacting protein* (*HHIP*), and *Inter-alpha-trypsin inhibitor heavy chain H1* (*ITIH1*). Within Finger OA, two proteins were implicated: *Matrix Gla protein* (*MGP*) and *Ecto-ADP-ribosyltransferase 4* (*ART4*). Lastly, Spine OA and Thumb OA each identified one protein, *interleukin-12 subunit beta* (*IL12B*) and *repulsive guidance molecule B* (*RGMB*), respectively. OR greater than 1 in Table 2 indicates that an increase in the respective protein is associated with an increased risk of OA, while OR less than 1 suggests that an increase in the protein is associated with a decreased risk of OA.

## 3.3 Sensitivity analysis of the causal relationships between proteins and five OA phenotypes

Table 3 summarizes the main results of the reverse causal analysis, Bayesian colocalization analysis, and phenotype scan. Firstly, in the reverse MR analysis, except for IL12B, which yielded no results, the p-values calculated from the five methods for the other proteins were all greater than 0.05, indicating no causal relationship between OA and the respective screened proteins. Specifically, although one significant SNP was identified for Spine OA, it did not have a corresponding SNP in the IL12B dataset, preventing us from using the harmonise_data () function to merge the data. Due to the lack of matching SNPs, we were unable to conduct the reverse MR analysis for IL12B. The complete data is provided in S4 Table. Furthermore, the Steiger filtering provided further confirmation of our results. Secondly, the results of the Bayesian colocalization analysis showed that seven proteins met the criteria. Specifically, *DNAJB12* (PP.H4.abf = 0.902) and *USP8* (PP.H4.abf = 0.864) share the same causal variant

**Table 2. MR results of plasma proteins significantly associated with OA after Bonferroni correction.**

| Protein | OA | UniProt ID | SNP | Method | SE | OR(95% CI) | P |
|---------|-----|-----------|-----|--------|-----|-----------|-----|
| *DNAJB12* | Knee | Q9NXW2 | rs9416018 | Wald ratio | 0.09 | 1.51(1.26,1.81) | 6.57E-06 |
| *USP8* | | P40818 | rs62019074 | Wald ratio | 0.10 | 1.72(1.42,2.08) | 2.24E-08 |
| COL2A1 | | P02458 | rs10875723 | Wald ratio | 0.01 | 0.95(0.93,0.97) | 1.46E-06 |
| ITGA2 | Hip | P17301 | rs181769 | Wald ratio | 0.11 | 0.64(0.52,0.78) | 2.19E-05 |
| HHIP | | Q96QV1 | rs11727676 | Wald ratio | 0.06 | 1.42(1.26,1.62) | 4.18E-08 |
| ITIH1 | | P19827 | rs1042779 | Wald ratio | 0.02 | 0.88(0.85,0.91) | 6.77E-13 |
| MGP | Finger | P08493 | rs7294636 | Wald ratio | 0.07 | 0.56(0.48,0.64) | 2.99E-16 |
| ART4 | | Q93070 | rs11276 | Wald ratio | 0.02 | 1.17(1.12,1.22) | 1.09E-14 |
| *IL12B* | Spine | P29460 | rs10043720 | Wald ratio | 0.03 | 0.87(0.81,0.92) | 3.12E-06 |
| *RGMB* | Thumb | Q6NW40 | rs61048056 | Wald ratio | 0.12 | 0.59(0.47,0.75) | 1.12E-05 |

with Knee OA; *HHIP* (PP.H4.abf = 0.999) and *ITIH1* (PP.H4.abf = 0.979) share the same causal variant with Hip OA; *MGP* (PP.H4.abf = 0.966) with Finger OA; *IL12B* (PP.H4. abf = 0.978) with Spine OA; and *RGMB* (PP.H4.abf = 0.932) with Thumb OA.

According to the results of phenotype scanning, we found associations between rs11727676 and BMI, bilateral leg impedance, and height. Similarly, rs1042779 showed associations with AP-4 complex subunit mu-1 and Hip/Knee OA. Additionally, rs7294636 was associated with grip strength in both hands. Considering the potential horizontal pleiotropy, these significant SNPs may influence OA through their effects on the aforementioned phenotypes, rather than directly through plasma proteins themselves. Therefore, we decided to exclude *HHIP*, *ITIH1*, and *MGP* proteins from further analysis. On the other hand, for DNAJB12, USP8, RGMB, and IL12B, we observed no significant SNP associations with other traits; any observed associations were limited to their respective proteins (as shown in Tables 3 and S5). These results can be considered as the final conclusions of our study.

**Table 3. Reverse causal relationship detection, Bayesian co-localization analysis, and phenotype scanning results of 10 potential pathogenic proteins.**

| Protein | OA | SNP | Reverse MR ($P_{IVW}$) | Steiger filtering | PPH4.abf | Phenotype scan |
|---------|-----|-----|-----------------------|-------------------|----------|----------------|
| *DNAJB12* | Knee | rs9416018 | 0.982 | 2.46E-15 | 0.902 | N/A |
| *USP8* | | rs62019074 | 0.521 | 2.28E-13 | 0.864 | N/A |
| COL2A1 | | rs10875723 | 0.207 | 8.15E-272 | 0.02 | Collagen alpha-1(I) chain |
| ITGA2 | Hip | rs181769 | 0.480 | 1.65E-20 | 0.662 | N/A |
| HHIP | | rs11727676 | 0.300 | 4.24E-52 | 0.999 | BMI、 Comparative height size at age 10、 Impedance of leg left、 |
| ITIH1 | | rs1042779 | 0.848 | 3.25E-257 | 0.979 | AP-4 complex subunit mu-1、 Arylamine N-acetyltransferase 1、 Hip or knee osteoarthritis |
| MGP | Finger | rs7294636 | 0.144 | 1.22E-137 | 0.966 | Ecto-ADP-ribosyltransferase 4、 Uncarboxylated MGP、 Left and right hand grip strength |
| ART4 | | rs11276 | 0.062 | 1.87E-242 | 0.563 | |
| *IL12B* | Spine | rs10043720 | — | 1.23E-266 | 0.978 | Interleukin-23、 Crohn's disease、 Inflammatory bowel disease、 Ulcerative colitis、 Self-reported psoriasis、 |
| *RGMB* | Thumb | rs61048056 | 0.969 | 4.52E-48 | 0.932 | RGM domain family member B |

### 3.4 Cross-trait LDSC analysis of sample overlap between proteins and OA

We used the cross-trait LDSC method to assess the genetic relationship between four major proteins and OA, with a particular focus on the potential impact of sample overlap on genetic correlation estimates (see Table 4). The results show that for Knee OA, the intercepts for DNAJB12 (0.0073, SE = 0.0053) and USP8 (-0.0047, SE = 0.0053) were both close to zero, with p-values greater than 0.05, indicating that the difference between these intercepts and zero was not statistically significant. This suggests that there is no significant sample overlap between DNAJB12, USP8, and Knee OA, and supports the notion that the genetic correlation estimates between these proteins and Knee OA were not influenced by sample overlap.Similarly, for IL12B (intercept = -0.0067, SE = 0.0053) with Spine OA and RGMB (intercept = 0.0033, SE = 0.0056) with Thumb OA, the analysis showed no significant sample overlap between the proteins and the corresponding phenotypes. This further suggests that the genetic correlation estimates for IL12B and RGMB with Spine OA and Thumb OA were not disturbed by sample overlap.

### 3.5 External validation

For external validation, we conducted analyses using different variation and significance strategies within the same dataset. The results showed consistency with the initial analysis for Finger OA, SpineOA, and Thumb OA (Table 5). However, due to the lack of proteins such as *DNAJB12* and *USP8* in the new plasma pQTLs, we were unable to calculate their causal relationship with OA, which is attributed to the diversity of the data. Nevertheless, we believe that the results of external validation still serve as credible evidence for the identification of the plasma proteins.The complete data is provided in S6 Table.

### 3.6 Screening and molecular docking of small molecule drugs

After a series of analyses and identifications, we finally identified four proteins: *DNAJB12*, *USP8*, *RGMB* and *IL12B*. Small molecule drugs targeting these four proteins were searched in the DsigDB database, and the detailed results are shown in S7 Table. Based on the adjusted P-value < 0.05 and the combined score, 3 drugs were identified for *DNAJB12* and *USP8*, respectively, and 10 drugs were identified for *IL12B*, whereas there were no drugs that met the conditions for *RGMB*.

After obtaining the 3-dimensional structures of the proteins and molecules by the above method, the binding energies under the combination between them were calculated and the results are shown in Fig 2A. The binding energies of *DNAJB12* with Doxazosin, XEN 103 with *USP8*, Montelukast, IB-Meca, AY_9944, Phorbol_12,13-dibutyrate with *IL12B* binding energy all meet the threshold and are our final predicted small molecule drugs. Fig 2B–2D are visualisations of the patterns in each protein that bind best to the ligand molecule.

**Table 4. Summary of genetic correlation results.**

| Proteins | OA | rg | rg_se | rg_P | Intercept | Intercept_se | Intercept_P |
|----------|-----|------|-------|------|-----------|--------------|-------------|
| DNAJB12 | Knee | 0.0822 | 0.0650 | 0.2065 | 0.0073 | 0.0053 | 0.166 |
| USP8 | Knee | 0.0778 | 0.0900 | 0.3874 | -0.0047 | 0.0053 | 0.372 |
| IL12B | Spine | 0.0021 | 0.0726 | 0.9765 | -0.0067 | 0.0053 | 0.208 |
| RGMB | Thumb | -0.2258 | 0.0872 | 0.1254 | 0.0033 | 0.0056 | 0.555 |

**Table 5. External validation results using new plasma pQTL data.**

| Protein | OA | UniProt ID | SNP | Method | OR(95%CI) | SE | P |
|---------|-----|-----------|------|--------|-----------|-----|-----|
| MGP | finger | P08493 | rs7135211 | Wald ratio | 0.577(0.503, 0.662) | 0.07 | 3.73E-16 |
| ART4 | | Q93070 | rs1001096 | Wald ratio | 1.209(1.163, 1.258) | 0.02 | 1.02E-14 |
| SPON2 | knee | Q9BUD6 | rs878323 | Wald ratio | 0.835(0.772, 0.903) | 0.04 | 6.02E-07 |
| *IL12B* | spine | P29460 | rs4921484 | Wald ratio | 0.869(0.820, 0.922) | 0.03 | 4.35E-05 |
| *RGMB* | thumb | Q6NW40 | rs1563317 | Wald ratio | 0.712(0.608, 0.833) | 0.08 | 1.40E-05 |

## 4 Discussion

We employed a comprehensive analytical approach, integrating MR and colocalization techniques, to assess proteins associated with the pathogenesis of OA[39]. The objective is to identify biomarkers that can be utilized for early diagnosis and risk assessment. We acknowledge that the "causal relationships" identified through MR may be influenced by various factors, including reverse causality, horizontal pleiotropy, or genetic confounding [20]. Therefore, to minimize bias, we conducted bidirectional MR analyses, and the proteins identified in the preliminary MR analysis did not exhibit reverse causality. This conclusion was further supported by the Steiger filtering method [40]. Additionally, to limit bias introduced by horizontal pleiotropy, we only utilized cis-pQTLs as instruments, as they directly act in the transcription or translation processes [41].

In addition, we also employed Bayesian co-localization methods to eliminate biases caused by linkage disequilibrium. By setting a posterior probability threshold of 0.8, we identified seven proteins (*DNAJB12*, *USP8*, *HHIP*, *ITIH1*, *MGP*, *IL12B*, *RGMB*). Further phenotype scans revealed that SNPs associated with *HHIP*, *ITIH1*, *MGP*, *IL12B*, and *RGMB* were correlated with other traits. Specifically, a significant SNP (rs11727676) in *HHIP* was found to be associated with BMI and impedance in the lower limbs. These may act as confounding factors for OA, suggesting that rs11727676 may not have a causal relationship with OA through plasma protein but rather mediated by these confounding factors. Similarly, Hip or knee

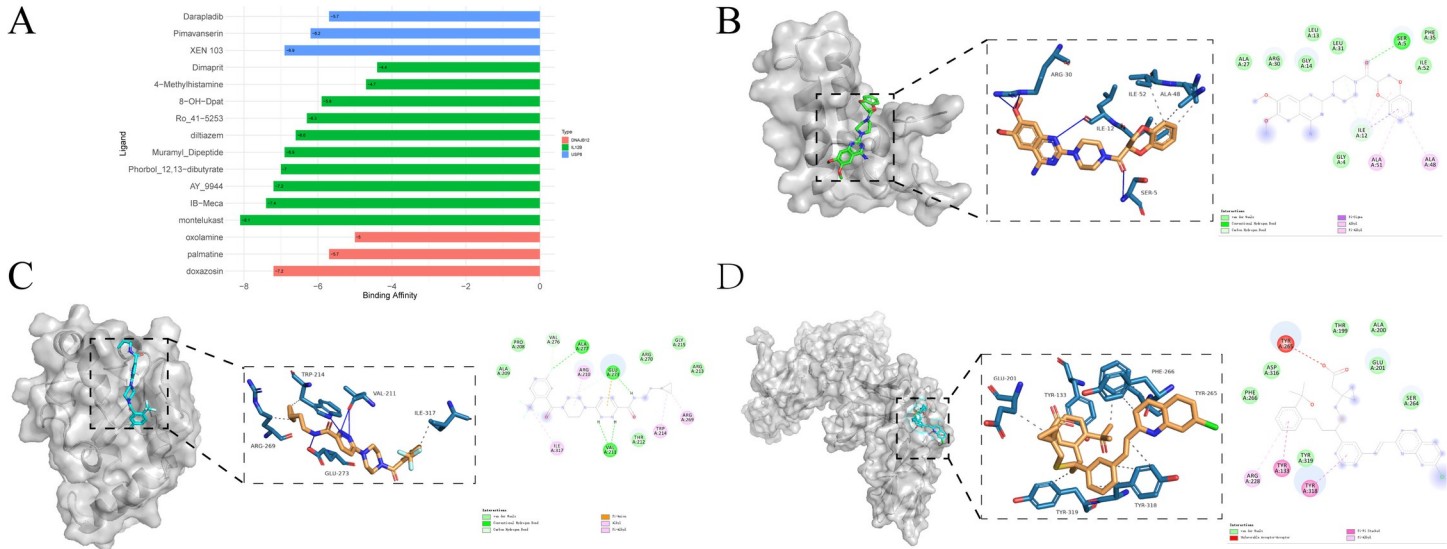

**Fig 2. Visualization of molecular docking results.** (A) Binding affinity of proteins to small molecule drugs, (B) 3D and 2D visualisation of DNAJB12 docking with Doxazosin, (C) 3D and 2D visualisation of USP8 docking with XEN 103, (D) 3D and 2D visualisation of IL12B docking with Montelukast.

osteoarthritis associated with rs1042779 and grip strength associated with rs7294636 were considered confounding factors. Regarding rs10043720, although its association with Crohn's disease [42], Inflammatory bowel disease [43], and their co-occurrence with SpA [43] has been reported in the literature, the relationship between them is still unclear, thus unable to fully explain the relationship between the identified proteins and OA. Finally, we successfully determined causal relationships between four proteins and three types of OA. *DNAJB12* and *USP8* were considered risk factors for Knee OA, *RGMB* was identified as a protective factor for ThumbOA, and *IL12B* was considered a protective factor for SpineOA.

*DNAJB12*, a member of the Hsp40 protein family, is intricately linked to the function of the endoplasmic reticulum (ER). Its primary role involves guiding Hsp70 proteins to the surface of the ER and coordinating the interaction between the ER and cytoplasmic chaperone systems in protein quality control [44]. The ER itself performs a multitude of biological functions, including regulating protein biosynthesis, facilitating the correct folding of oxidized proteins, secretion, maintaining protein quality, calcium signal transduction, autophagy, stress response, and apoptosis. Research indicates that the ER transmembrane Hsp40 co-partner, *DNAJB12*, in conjunction with HSPA/Hsp70, specifically identifies and processes misfolded membrane proteins, thereby participating in ER-associated autophagy (ERAA) [45]. ERAA is a crucial protein degradation pathway essential for maintaining intracellular protein balance and eliminating aberrant proteins and damaged organelles. Although current research has not directly linked *DNAJB12* with Knee OA, further exploration of *DNAJB12*'s mechanism in ER autophagy could reveal its potential role in diseases like Knee OA.

*USP8*, part of the deubiquitinating enzyme (Dubs) family, plays a key role in intracellular protein degradation and signal transduction processes. Studies have shown that ubiquitination is a dynamic process reversible by Dubs [46,47]. Particularly, *USP8* is instrumental in modulating the Smo protein within the Hedgehog (Hh) signaling pathway. Experimental evidence from fruit fly wing disks has demonstrated that loss of *USP8* function diminishes the intracellular accumulation of Smo under Hh stimulation, whereas overexpression of *USP8* leads to abnormal accumulation of Smo and overactivation of the Hh pathway [48,49]. Similarly, in S2 cells, overexpression of *USP8* reduces the ubiquitination of Smo, while downregulation of *USP8* weakens the Hh-induced ubiquitin-mediated degradation of Smo. Additionally, Hh regulates *USP8* through a PKA/CK1-independent mechanism [48,49]. This suggests that *USP8*, as a regulator of the Hh signaling pathway, impacts the activation of this pathway through its involvement in the deubiquitination of Smo protein. The activation of the Hh pathway has been shown to be significant in the progression of OA [50]. Therefore, our research supports the notion that *USP8* may influence the onset and progression of osteoarthritis. This potential causal relationship warrants further in-depth investigation.

We found that elevated levels of *IL12B* and *RGMB* have a protective effect on the occurrence of corresponding OA. As a member of The repulsive guidance molecule (RGM) proteins family, *RGMB* serves as an auxiliary receptor for bone morphogenetic proteins (BMP) and a sensitizer for BMP signaling [51]. The transforming growth factor-β (TGFβ) superfamily ligands, especially BMP, guide skeletal patterning, chondrogenesis, and osteogenesis in vertebrate development, as well as serve as guidance cues for neuronal lineage commitment and promote neuronal differentiation in the nervous system [52,53]. Studies have revealed that *RGMB* enhances BMP signaling by directly binding to BMP-2 and BMP-4, followed by binding to type I and type II BMP receptors, and both *RGMB* and BMP signaling have a positive regulatory effect on axon extension in vitro and early axonal regeneration after neural injury in vivo [51,54].

Research on the role of *RGMB* in ThumbOA has not been reported to date, but we infer two reasons why *RGMB* may serve as a therapeutic target: First, osteoarthritis is usually

accompanied by vascular invasion, and neurogenesis and vasculature are linked through common pathways [55]. Due to the release of inflammatory factors and changes in the local environment, these newly formed blood vessels may affect the surrounding neural tissues. Second, ThumbOA and nerve injury may share some common pathophysiological mechanisms, such as inflammatory responses and local circulatory disturbances, which may simultaneously affect both the joints and the nervous system.

*IL12B* is a subunit of the heterodimeric cytokines IL-12 and IL-23 [56], both of which play important roles in host defense against pathogens and wound healing[57]. Previous research by Weronica E and others [58] investigated the causal effects of inflammatory proteins on inflammatory diseases and found that *IL12B* has a protective effect in psoriatic arthritis. Despite this, it contradicts the therapeutic purpose of *IL12B*-targeting monoclonal antibody ustekinumab in treating psoriasis and psoriatic arthritis by reducing *IL12B* signaling [58]. They stated that the beneficial effects of these cytokines in protecting healthy tissues from damage are significantly different from the effects of excessive cytokine production in chronic inflammatory conditions such as psoriasis and psoriatic arthritis. Similar to the nature of our research findings, since the study of *IL12B* in spine-related diseases is in its infancy, further clinical research and experiments are needed to validate these results.

Our drug prediction results provide several drugs, including Doxazosin, an alpha-1 antagonist with a wide range of applications in anti-tumour and hypertension; XEN 103 and AY_9944 are less well studied and the mechanism of action is unknown; Montelukast belongs to the leukotriene receptor antagonist (LTRA) class of drugs and is used to treat asthma; IB-Meca is an anti-inflammatory drug used to treat patients with rheumatoid arthritis. Among the available studies, we did not find any that reported on the role of these drugs in OA. However, by performing docking analyses of the proteins involved in OA with these drugs, we observed some encouraging results. In particular, we found that these drugs are able to act on causal proteins, which may lead to an effect on OA symptoms. Overall, although further experimental and clinical studies are needed to validate these findings, these docking results provide useful clues for future directions in drug discovery and treatment of OA, and further studies will help to unravel the association between these drugs and the pathophysiological mechanisms of OA, and may provide important insights for the development of new therapeutic strategies.

## 5 Limitations

Our study has several limitations. First, we examined the effects of proteins using data from different studies, where varying measurement methodologies may lead to biased results. Specifically, discrepancies in measurement techniques across these studies could compromise the comparability and interpretability of our findings. However, it is noteworthy that the circular proteomic data utilized in GWAS studies by Ferkingstad [19], Zheng [20], and others were all based on aptamers, providing a degree of methodological consistency.

Second, our analysis was confined to populations of European descent, limiting the generalizability of our results to other ethnic groups. The inclusion of diverse ethnic populations is crucial for a comprehensive understanding of OA's epidemiology on a global scale, and future research should aim to encompass these varied demographics.

Lastly, although we have identified several pathogenic proteins associated with OA and made drug predictions, the results are only a starting point. Further validation and in-depth biological analyses are essential to determine the exact role of these proteins in the pathogenesis of OA. This will require subsequent experimental studies to confirm our findings and translate them effectively into clinical applications, especially in non-European populations.

## 6 Conclusions

Our study ultimately identified four circulating proteins with significant causal relationships, corresponding to three types of OA, providing valuable insights for drug prediction. These findings not only aid in more precise early diagnosis and risk assessment but also offer new potential drug targets. This paves the way for developing treatment strategies tailored to various types of osteoarthritis.

## Supporting information

**S1 Table. STROBE-MR-checklist-fillable.**
(DOCX)

**S2 Table. Significant instrumental variables for circulating proteins and OA.**
(XLSX)

**S3 Table. MR results of circulating proteins and five OA.**
(XLSX)

**S4 Table. Reverse causality test for circulating proteins and five OA.**
(XLSX)

**S5 Table. Phenotype scanning results.**
(XLSX)

**S6 Table. Results of external validation.**
(XLSX)

**S7 Table. Small molecule drug forecasting.**
(XLSX)

## Acknowledgments

We thank the authors of the studies by Ferkingstad et al. and Zheng et al. for providing the plasma protein quantitative trait loci data, and the researchers involved in the GWAS meta-analysis on osteoarthritis for the summary statistics used in this study.

## Author Contributions

**Conceptualization:** Yanan Xu, Shuai Chen, Li Guo, Xiaochun Wei.

**Data curation:** Chengyang Lu, Yanan Xu, Shuai Chen.

**Formal analysis:** Chengyang Lu.

**Funding acquisition:** Pengcui Li, Xiaochun Wei, Xueqin Rong.

**Investigation:** Yanan Xu, Li Guo.

**Methodology:** Chengyang Lu, Yanan Xu.

**Resources:** Li Guo.

**Software:** Chengyang Lu, Yanan Xu, Shuai Chen.

**Supervision:** Li Guo, Pengcui Li, Xiaochun Wei, Xueqin Rong.

**Validation:** Yanan Xu, Li Guo, Pengcui Li, Xueqin Rong.

**Writing – original draft:** Chengyang Lu.

**Writing – review & editing:** Chengyang Lu, Yanan Xu, Shuai Chen, Li Guo, Pengcui Li, Xiaochun Wei, Xueqin Rong.

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
