## [Decision Letter · Decision Letter 0]

7 Oct 2024

PONE-D-24-36317Mendelian randomisation analysis to identify potential drug targets for osteoarthritisPLOS ONE

Dear Dr.  Wei,

Thank you for submitting your manuscript to PLOS ONE. After careful consideration, we feel that it has merit but does not fully meet PLOS ONE’s publication criteria as it currently stands. Therefore, we invite you to submit a revised version of the manuscript that addresses the points raised during the review process.1) Both reviewers were positive about your manuscript. They both indicate that your results support your conclusions and that your statistical analyses were appropriate. Each reviewer provided a document with comments that you should address to improve the readability of the manuscript and to clarify numerous points. These two documents should be attached to this decision letter. If they are not attached, email PLOS One staff.2) Reviewer #1 points: #6,7,10,15--please provide detailed explanations, rationales, as requested. #2, asks for a complete STROBE-MR checklist. The points not mentioned  (many of which refer to formatting issues) should also be addressed.3) Reviewer #2: thirteen comments are given. Comment #2 asks about possible sample overlap between the protein and OA GWAS. Comment #3 suggests checking why the F statistic is still needed for IV selection. I am not going to paraphrase all of the points. The points made by this reviewer are excellent. You should address the various points, doing so will greatly improve your manuscript. Please also address the comments under the heading "clarification questions."4) Addressing the points raised by the two reviewers will improve your manuscript.

We look forward to receiving your revised manuscript.

Kind regards,

Stephan N. Witt, Ph.D.

Academic Editor

PLOS ONE

Journal Requirements:

1. When submitting your revision, we need you to address these additional requirements. Please ensure that your manuscript meets PLOS ONE's style requirements, including those for file naming. The PLOS ONE style templates can be found at https://journals.plos.org/plosone/s/file?id=wjVg/PLOSOne_formatting_sample_main_body.pdf and https://journals.plos.org/plosone/s/file?id=ba62/PLOSOne_formatting_sample_title_authors_affiliations.pdf 2. Please note that PLOS ONE has spec6ific guidelines on code sharing for submissions in which author-generated code underpins the findings in the manuscript. In these cases, all author-generated code must be made available without restrictions upon publication of the work. Please review our guidelines at https://journals.plos.org/plosone/s/materials-and-software-sharing#loc-sharing-code and ensure that your code is shared in a way that follows best practice and facilitates reproducibility and reuse. 3. Thank you for stating the following financial disclosure: "This manuscript was supported by funding from grants National Natural Science Foundation of China U23A6009, U21A20353, 82172503, Natural Science Foundation of Shanxi Province 20210302123285, Key R&D Program Projects of Shanxi Province 202202040201012, Hainan Provincial Medical and Health Research Program 21A200349."  Please state what role the funders took in the study.  If the funders had no role, please state: "The funders had no role in study design, data collection and analysis, decision to publish, or preparation of the manuscript." If this statement is not correct you must amend it as needed. Please include this amended Role of Funder statement in your cover letter; we will change the online submission form on your behalf. 4. Thank you for uploading your study's underlying data set. Unfortunately, the repository you have noted in your Data Availability statement does not qualify as an acceptable data repository according to PLOS's standards. At this time, please upload the minimal data set necessary to replicate your study's findings to a stable, public repository (such as figshare or Dryad) and provide us with the relevant URLs, DOIs, or accession numbers that may be used to access these data. For a list of recommended repositories and additional information on PLOS standards for data deposition, please see https://journals.plos.org/plosone/s/recommended-repositories. 5. Please include captions for your Supporting Information files at the end of your manuscript, and update any in-text citations to match accordingly. Please see our Supporting Information guidelines for more information: http://journals.plos.org/plosone/s/supporting-information.

Reviewers' comments:

Reviewer's Responses to Questions

**Comments to the Author**

1. Is the manuscript technically sound, and do the data support the conclusions?

Reviewer #1: Yes

Reviewer #2: Yes

2. Has the statistical analysis been performed appropriately and rigorously? 

Reviewer #1: Yes

Reviewer #2: Yes

3. Have the authors made all data underlying the findings in their manuscript fully available?

Reviewer #1: Yes

Reviewer #2: Yes

4. Is the manuscript presented in an intelligible fashion and written in standard English?

Reviewer #1: Yes

Reviewer #2: Yes

5. Review Comments to the Author

Reviewer #1: I have reviewed the manuscript, and I find no concerns regarding dual publication, research ethics, or publication ethics. The authors have followed ethical guidelines appropriately in their research and reporting.

Reviewer #2: The authors conduct two-sample summary-data Mendelian randomization (MR) analyses to investigate the causal relationship between 1553 circulating proteins (as the exposure) and 5 osteoarthritis (OA) related traits (as the outcome). The causal effect of each protein on each OA trait is estimated using the Wald ratio (with a single variant) or inverse-variance weighted (IVW) approach (with multiple variants) as the main analysis and validated through a series of sensitivity checks, including reverse causality analyses, co-localization tests and phenotype searching. The research question is well-defined, and the estimation strategy is clearly outlined. The initial MR results are effectively validated and filtered. Following the statistical analyses, a detailed discussion of the underlying biological mechanisms and drug target implications are provided. My detailed comments are attached in the PDF file.

6. PLOS authors have the option to publish the peer review history of their article (what does this mean?). If published, this will include your full peer review and any attached files.

Reviewer #1: No

Reviewer #2: No

---

## [Author Response · Author response to Decision Letter 0]

5 Nov 2024

Dear Reviewer,

Thank you very much for your detailed review and valuable suggestions on my manuscript. Your feedback has been incredibly helpful and has allowed me to gain a deeper understanding of some key issues during the revision process. Although I have made every effort to address your comments, I realize there may still be areas that need further improvement.

If there are any aspects that remain insufficient, I would greatly appreciate your continued guidance, and I am more than willing to make additional revisions as needed. Thank you again for your support and your patience in reviewing my work!

I truly appreciate your valuable time and assistance.

---

## [Decision Letter · Decision Letter 1]

5 Dec 2024

PONE-D-24-36317R1Mendelian randomization analysis to identify potential drug targets for osteoarthritisPLOS ONE

Dear Dr. Wei,

Thank you for submitting your manuscript to PLOS ONE. After careful consideration, we feel that it has merit but does not fully meet PLOS ONE’s publication criteria as it currently stands. Therefore, we invite you to submit a revised version of the manuscript that addresses the points raised during the review process.

Reviewer #2 reviewed your revised manuscript and has some additional comments that need to be addressed. Addressing their comments will improve your paper. There are seven major points, and you will see that several of them are minor and can be addressed easily. Please read the points carefully and revise accordingly. Points 8 and 9 are about references and writing style, respectively. Please address these points too.

 Again, please know that addressing these new points will improve the quality of your study,

We look forward to receiving your revised manuscript.

Kind regards,

Stephan N. Witt, Ph.D.

Academic Editor

PLOS ONE

Journal Requirements:

Reviewers' comments:

Reviewer's Responses to Questions

**Comments to the Author**

1. If the authors have adequately addressed your comments raised in a previous round of review and you feel that this manuscript is now acceptable for publication, you may indicate that here to bypass the “Comments to the Author” section, enter your conflict of interest statement in the “Confidential to Editor” section, and submit your "Accept" recommendation.

Reviewer #1: All comments have been addressed

Reviewer #2: (No Response)

2. Is the manuscript technically sound, and do the data support the conclusions?

Reviewer #1: Yes

Reviewer #2: Yes

3. Has the statistical analysis been performed appropriately and rigorously? 

Reviewer #1: Yes

Reviewer #2: Yes

4. Have the authors made all data underlying the findings in their manuscript fully available?

Reviewer #1: Yes

Reviewer #2: Yes

5. Is the manuscript presented in an intelligible fashion and written in standard English?

Reviewer #1: Yes

Reviewer #2: Yes

6. Review Comments to the Author

Reviewer #1: The authors have provided detailed and thoughtful responses to my comments, demonstrating a strong commitment to improving the manuscript. Their revisions address the concerns I raised effectively, and the updated version of the manuscript is much clearer and more robust as a result.

I appreciate the effort the authors have put into explaining their methods and clarifying their results. The study presents valuable findings with significant implications for the field, and the methodology is well-executed and appropriately detailed. The results are supported by comprehensive statistical analysis and are presented in a clear and logical manner.

In my opinion, this work meets the high standards of PLOS ONE in terms of scientific rigor, transparency, and accessibility. Furthermore, the findings contribute meaningful insights that have the potential to impact future research in this area. I believe this manuscript is a valuable addition to the scientific literature and recommend it for publication.

Reviewer #2: The authors have effectively addressed the majority of my concerns and questions, but I have a few additional comments regarding their responses, which are listed in the PDF file.

7. PLOS authors have the option to publish the peer review history of their article (what does this mean?). If published, this will include your full peer review and any attached files.

Reviewer #1: No

Reviewer #2: No

---

## [Author Response · Author response to Decision Letter 1]

8 Dec 2024

Dear Editor and Reviewers,

Thank you for the opportunity to resubmit our manuscript titled “Mendelian randomization analysis to identify potential drug targets for osteoarthritis” (Manuscript ID: PONE-D-24-36317R1). We would like to express our sincere gratitude for the thoughtful and constructive feedback provided by you and the reviewers.

We are grateful to the first reviewer for agreeing with the revisions we made during the first round of revision and appreciate their valuable suggestions.

In response to the second reviewer’s additional seven comments, we have carefully addressed each of the points raised. Several of the suggestions were straightforward to implement, and we believe these revisions have significantly improved the quality of the manuscript. A detailed response to each of the second reviewer’s comments, along with explanations of the changes made, is included in the attached “Response to Reviewers” document.

We are confident that these revisions have fully addressed the second reviewer’s concerns and further strengthened the content of the paper. Please do not hesitate to contact us if any further clarification or modification is needed.

Once again, thank you for your time, consideration, and the reviewers’ insightful feedback.

---

## [Editor Report · Decision Letter 2]

17 Dec 2024

Mendelian randomization analysis to identify potential drug targets for osteoarthritis

PONE-D-24-36317R2

Dear Dr. Wei,

We’re pleased to inform you that your manuscript has been judged scientifically suitable for publication and will be formally accepted for publication once it meets all outstanding technical requirements.

Sorry for the delay in notifying you of my decision. 

Kind regards,

Stephan N. Witt, Ph.D.

Academic Editor

PLOS ONE
---

## [Editor Report · Acceptance letter]

20 Dec 2024

PONE-D-24-36317R2 

PLOS ONE

Dear Dr. Wei, 

I'm pleased to inform you that your manuscript has been deemed suitable for publication in PLOS ONE. Congratulations! Your manuscript is now being handed over to our production team.

Kind regards, 

on behalf of

Dr. Stephan N. Witt 

Academic Editor

PLOS ONE